# Hydrogel-embedded vertically aligned metal-organic framework nanosheet membrane for efficient water harvesting

Lingyue Zhang[1], Ruiying Li[1], Shuang Zheng [1], Hai Zhu[1], Moyuan Cao[2], Mingchun Li[3], Yaowen Hu[1], Li Long[1], Haopeng Feng[1] & Chuyang Y. Tang [1] ✉

Highly porous metal-organic framework (MOF) nanosheets have shown promising potential for efficient water sorption kinetics in atmospheric water harvesting (AWH) systems. However, the water uptake of single-component MOF absorbents remains limited due to their low water retention. To overcome this limitation, we present a strategy for fabricating vertically aligned MOF nanosheets on hydrogel membrane substrates (MOF-CT/PVA) to achieve ultrafast AWH with high water uptake. By employing directional growth of MOF nanosheets, we successfully create superhydrophilic MOF coating layer and pore channels for efficient water transportation to the crosslinked flexible hydrogel membrane. The designed composite water harvester exhibits ultrafast sorption kinetics, achieving 91.4% saturation within 15 min. Moreover, MOF-CT/PVA exhibits superior solar-driven water capture-release capacity even after 10 cycles of reuse. This construction approach significantly enhances the water vapor adsorption, offering a potential solution for the design of composite MOF-membrane harvesters to mitigate the freshwater crisis.

Water scarcity is a pressing global environmental challenge that is further exacerbated by climate change and population growth[1]. In recent years, atmospheric water harvesting (AWH) has emerged as a promising decentralized approach to water production[2,3], harnessing water vapor from the air across a wide range of relative humidity (RH) levels[4]. Unlike traditional centralized water purification methods reliant on waterbodies[5,6], AWH offers a viable solution for small-scale water production, particularly in low-populated and arid regions[7,8].

Metal-organic framework (MOF) materials have gained significant attention due to their exceptional properties such as high porosity, structural tunability, and stability[9,10]. As adsorbents, MOFs have shown great potential, with water-stable MOFs like MIL-101(Cr) demonstrating favorable performance in AWH by minimizing undesired sorption hysteresis[11]. However, their limited water uptake capabilities pose a significant challenge to their wider applications[7]. To overcome this

limitation, researchers have explored the incorporation of MOFs into polymeric substrates, particularly hydrogels, to create composite harvesters that enhance water vapor capture capacity and adsorption kinetics in AWH[12].

Despite the promising combination of MOFs with hydrogels, previous studies have faced challenges related to the dissolution or degradation of polymeric membranes when exposed to solvent systems containing polar aprotic agents like Dimethylformamide (DMF)[13], which coordinate with the metal centers in MOFs[14]. This necessitates the synthesis of MOF nanoparticles that are subsequently filtered onto substrate membranes[15]. Additionally, the exfoliation of MOF nanosheets, which offer hierarchically porous pathways and increased surface area for continuous AWH processes[16], typically requires ultrasonic treatment followed by random deposition onto the membrane surface[17]. These traditional exfoliation and doping methods often lead to significant bulk MOF residues[18], disordered packing, and

[1]Department of Civil Engineering, The University of Hong Kong, Hong Kong SAR, China. [2]School of Materials Science and Engineering, Smart Sensing Interdisciplinary Science Center, Nankai University, Tianjin, China. [3]School of Environment, Tsinghua University, Beijing, China. ✉e-mail: tangc@hku.hk

weak binding between the MOFs and substrate membranes[15]. In comparison to the controlled growth of MOF nanosheets, these drawbacks further restrict water uptake and decrease the lifespan of the MOF layer due to longitudinal transmission resistance and potential detachment[14].

To address these challenges, we propose a strategy to fabricate vertically aligned MOF nanosheets on a hydrogel membrane harvester (MOF-CT/PVA), exhibiting both superhydrophilicity and ultrafast water transport capacity. Polyethylene glycol (PEG) is incorporated as both a crosslinking and surfactant agent during the fabrication of the hydrogel membrane and the controlled growth of MOFs, respectively. This facile process enables the deliberate regulation of a vertically aligned nanosheet layer with robust flexibility and a high specific surface area for efficient water vapor adsorption. Moreover, the structural design of MOF-CT/PVA allows for solar-driven water release based on the solar-thermal properties of MOF nanosheets and morphology regulation. This study presents an innovative solution for the directional growth of MOF nanosheets on polymer substrates and showcases the application potential of such composite membrane harvesters in AWH.

## Results

### Creation of MOF nanosheet layer on hydrogel membranes

Chitosan (CT) and polyvinyl alcohol (PVA) are commonly used polymer hydrogel materials known for their low toxicity, high mechanical strength, and hydrophilicity[19]. Herein, PEG was initially employed as a crosslinker in the gel precursor (Supplementary Table 1), which was then blade-coated to achieve a controlled thickness before undergoing phase inversion to form the CT/PVA hydrogel substrate, as illustrated in Fig. 1a. The X-ray photoelectron spectroscopy (XPS) spectra confirmed that the chemically crosslinked CT/PVA substrate exhibited an increased relative content of C–O/C–N and C=O functional groups (Supplementary Fig. 1). This observation is consistent with the cross-linking reaction between PEG and CT/PVA[20,21]. These crosslinking points play a crucial role in maintaining the stability and integrity of the hydrogel substrate during the growth of Zn-TCPP (Fig. 1b). In contrast, the pristine substrate (without PEG) showed significant dissolution[22] and the formation of unzipping pores in the solvent environment (DMF and ethanol) of MOF synthesis (Supplementary Fig. 2).

In addition, the inclusion of PEG molecules, which possess alternating hydrophobic and hydrophilic groups, serves as non-ionic

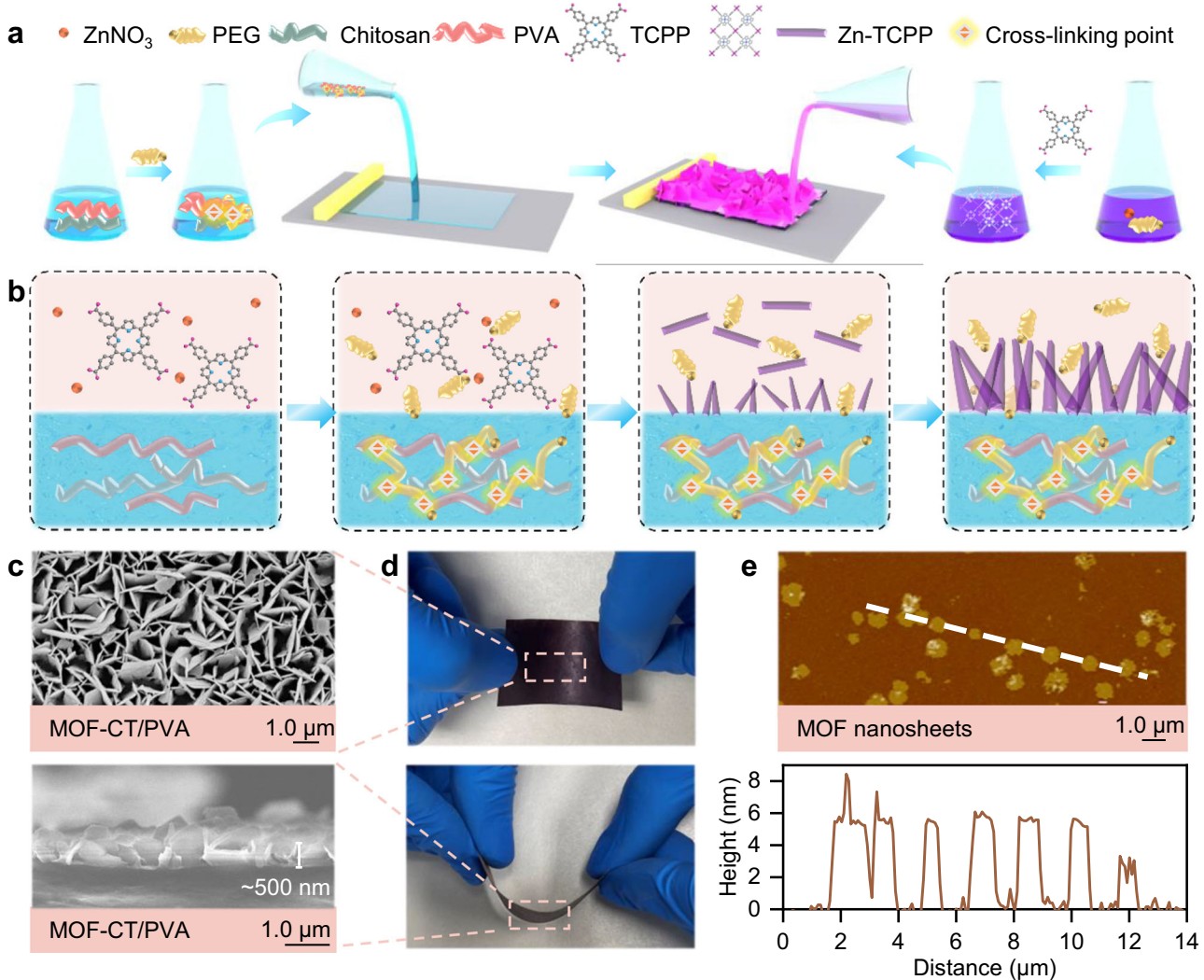

**Fig. 1 | Fabrication of vertically aligned MOF nanosheet layer on hydrogel membranes. a** Schematic of the membrane fabrication process. Hydrogel solution was blade-coated to form the CT/PVA substrate and immersed into the precursors for the directional growth of MOF nanosheets. **b** Schematic of the dual effects of PEG in the synthesis of MOF-CT/PVA, acting as both a crosslinking (hydrogel phase, blue) and surfactant agent (MOF phase, pink). PEG selectively attaches to the nanocrystals during the anisotropic growth of Zn-TCPP. **c** Vertically aligned nanostructure on the surface and cross-section of MOF-CT/PVA after bending and flipping tests. **d** Rigorous bending and flipping tests of MOF-CT/PVA. **e** AFM images and thickness of the MOF nanosheets (white dash line) detached from MOF-CT/PVA.

surfactants (Fig. 1b) to control the anisotropic growth of Zn-TCPP (TCPP = tetrakis(4-carboxyphenyl)porphyrin) by selectively attaching themselves to the nanocrystals[16]. Without PEG, isotropic growth results in the bulk crystal of Zn-TCPP, as shown in Supplementary Fig. 3. During the synthesis process, these surfactants adsorb at the nanocrystals and the interfaces of the aqueous (hydrogel) and organic (DMF and ethanol solution system used in Zn-TCPP fabrication) phases[23]. The directional distribution of PEG molecules incorporates crosslinking reactions between PEG and CT/PVA substrate, resulting in the formation of an oriented nanosheet layer on the hydrogel surface (Fig. 1a, c). Scanning electron microscope (SEM) images reveal a continuous petal-like MOF layer covering the hydrogel membrane substrate, wherein vertically aligned nanosheets possessing a thickness of less than 10 nm with reasonable uniformity (Fig. 1c). These results are consistent with the atomic force microscopy (AFM) images of the detached Zn-TCPP nanosheets (Fig. 1e). Furthermore, the presence of PEG in a directed arrangement at the interface of CT/PVA membrane substrate further influences the rectangular shape of Zn-TCPP compared to PEG-assisted Zn-TCPP powder materials[24]. In the absence of the aqueous membrane phase during the Zn-TCPP synthesis process, we observed disc-like nanoparticles with a multilayer thickness of 100 nm (Supplementary Fig. 3).

This designed MOF-CT/PVA membrane, benefiting from the dual effects of PEG in both aqueous and organic synthesis systems, exhibited remarkable flexibility and robustness. The morphology of the MOF layer demonstrated remarkable stability throughout rigorous bending and flipping tests (Fig. 1d), indicating that the nanosheets maintained a steady growth on the hydrogel membrane through interactions with PEG during the deformation process.

## Vertically aligned MOF nanosheets with superhydrophilicity
As shown in Supplementary Fig. 4, we observe the morphological characteristics of the MOF layer at different time intervals. In contrast to the previously reported horizontal stacking of nanosheets, a continuous and uniformly structured vertically aligned MOF-CT/PVA membrane gradually fabricated within 24 h. We hypothesized that this growth occurred in the out-of-plane orientation due to the attachment of PEG after nucleation and the increased resistance in the in-plane orientation[25]. To validate our hypothesis and gain further insight into the role of the surfactant, we produced non-modified bulk Zn-TCPP and detached the Zn-TCPP nanosheets from the MOF-CT/PVA.

The crystal structure and surface characteristics of the obtained materials were characterized. Fourier transform infrared (FTIR) spectra (Fig. 2a) demonstrated the presence of asymmetric and symmetric vibrational stretching of $COO^-$ was illustrated at 1604 cm$^{-1}$ and 1440 cm$^{-1}$, respectively, in Zn-TCPP nanosheets, MOF-CT/PVA membrane, and bulk Zn-TCPP, suggesting the construction of similar metalloporphyrin composition. The metalation of porphyrin by $Zn^{2+}$ ions was further confirmed through UV-vis absorption spectroscopy (Supplementary Fig. 5). Additionally, the stretching vibration of the ν (C−O) in Zn-TCPP nanosheets observed at 1255 cm$^{-1}$ after morphology regulation, indicating a significant interaction between the C−O group in PEG and the Zn-TCPP nanosheets (Supplementary Fig. 6). In

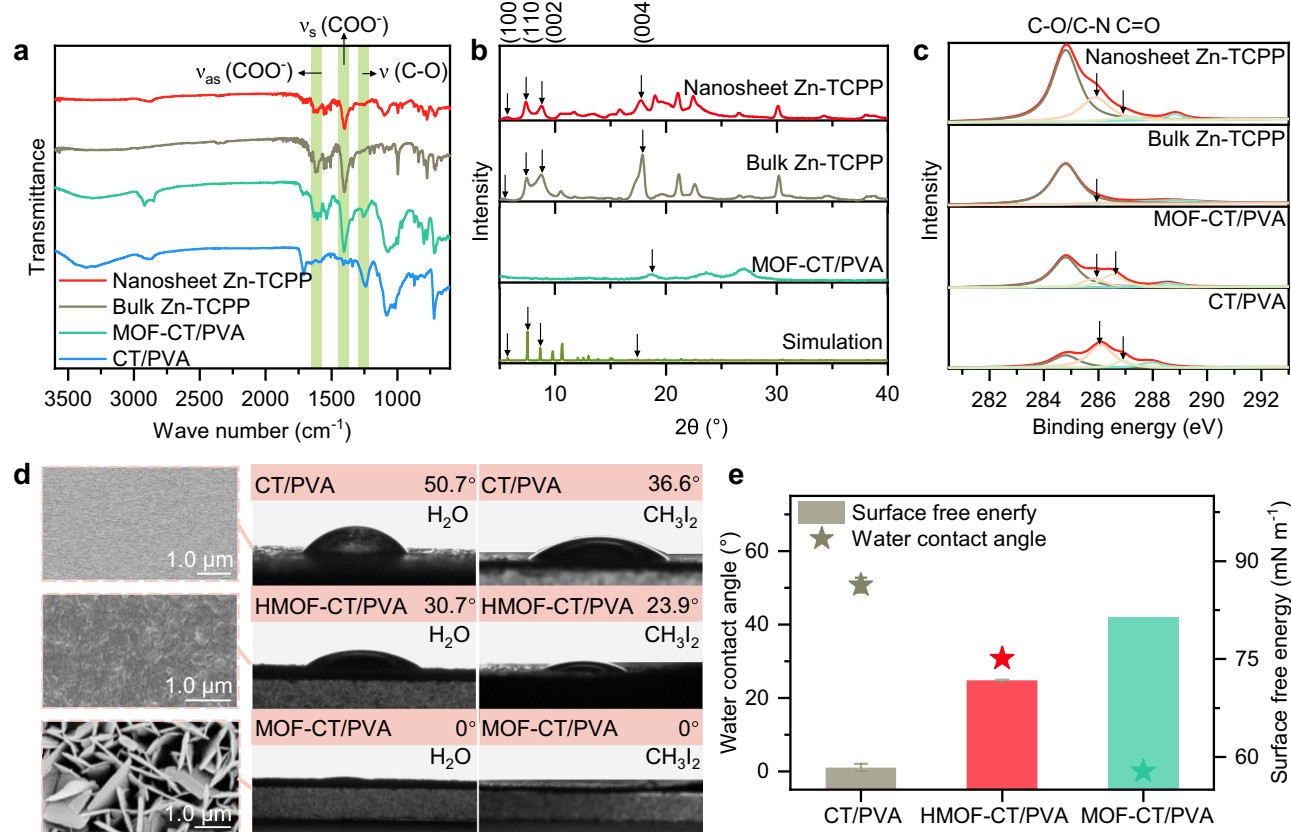

**Fig. 2 | Physicochemical properties of MOF nanosheets and hydrogel membranes. a** FTIR spectra of nanosheet Zn-TCPP, bulk Zn-TCPP, MOF-CT/PVA, and CT/PVA. **b** XRD patterns of nanosheet Zn-TCPP, bulk Zn-TCPP, MOF-CT/PVA, and simulated Zn-TCPP. **c** The C 1s spectra of XPS characterization of nanosheet Zn-TCPP, bulk Zn-TCPP, MOF-CT/PVA, and CT/PVA. **d** Morphology and surface hydrophilicity of CT/PVA, HMOF-CT/PVA, and MOF-CT/PVA. Zn-TCPP nanosheets were filtered onto the CT/PVA substrate to fabricate HMOF-CT/PVA. Inset: surface SEM images of CT/PVA, HMOF-CT/PVA, and MOF-CT/PVA (with pink dash frames). **e** Surface free energy and water contact angle of CT/PVA (gray), HMOF-CT/PVA (red), and MOF-CT/PVA (green). The error bars represent the standard deviation of data from three distinct samples (n = 3).

contrast, the stretching vibration of the ν (C−O) bond in bulk Zn-TCPP MOFs was very weak.

The X-ray diffraction (XRD) spectra (Fig. 2b) revealed that the detached Zn-TCPP nanosheets exhibited four characteristic peaks, displaying a similar pattern to that of bulk Zn-TCPP. This observation indicates the preservation of the identical tetragonal structure of Zn-TCPP in the nanosheet morphology. Due to the preferred parallel orientation of the 2D nanosheets on the hydrogel substrate, only a broad diffraction peak corresponding to the [004] plane was observed, confirming the crystal nature of the Zn-TCPP nanosheets on MOF-CT/PVA. High-resolution transmission electron microscopy images (Supplementary Fig. 7) further demonstrated the surface crystallinity of the Zn-TCPP nanosheets and exhibited lattice fringes with interplanar spacings measuring 1.32 nm, corresponding to the [110] plane[16]. Theoretical structure and Barrett−Joyner−Halenda (BJH) equations corroborate the measured data and confirm the layered structure (Supplementary Figs. 8 and 9). As the morphology of Zn-TCPP transitioned from three-dimensional to two-dimensional, the calculated layer spacing increased from 0.49 nm to 0.50 nm, as determined by XRD spectra and the Bragg equation[26]. The increase in layer spacing suggests the presence of surfactant interactions between the layers of the Zn-TCPP nanosheets. This surfactant attachment was further substantiated by the remarkable increase in the relative content of C−O/C−N functional groups observed in the C 1$s$ spectrum obtained from XPS analysis of the Zn-TCPP nanosheets (Fig. 2c, Supplementary Fig. 10 and Tables 2, 3). The presence of the C=O functional group in Zn-TCPP nanosheets was observed and provided evidence of a crosslinking reaction between the attached PEG and CT/PVA hydrogel.

This tetragonal crystal structure and incorporation of polar functional groups enable the MOF-CT/PVA membrane to exhibit extreme water-wetting property, resulting in a nearly zero water contact angle (Fig. 2d). To further compare directional MOF growth and conventional filtration of exfoliated MOF nanosheets, Zn-TCPP nanosheets were filtered onto the CT/PVA substrate to create a horizontally stacked MOF membrane (HMOF-CT/PVA) with the same mass fraction. It is important to note that despite the ultrasonic dispersion of the solution system of HMOF-CT/PVA beforehand, agglomeration and stacking phenomena unavoidably occur during the nanosheet filtration process. SEM images illustrate the stacking of nanosheets, resulting in the formation of a discontinuous layer with an in-plane orientation (Fig. 2d and Supplementary Fig. 11). The surface free energy of the CT/PVA substrate was calculated using the Owens, Wendt, Rabel, and Kaelble method (Supplementary Eq. 2), revealing a significant increase upon the application of the MOF coating. This enhancement elevated the surface free energy to nearly 81.4 mN m$^{-1}$, accompanied by a distinct vertically aligned morphology (Fig. 2e). It is noteworthy that the water droplet exhibited gradual absorption by the CT/PVA substrate and HMOF-CT/PVA within 60 and 10 s, respectively, whereas MOF-CT/PVA took less than a second (Supplementary Fig. 12 and Supplementary Movie 1). This observation highlights the superior hydrophilic properties of the MOF-CT/PVA and its potential for various applications requiring rapid water absorption. Since horizontal Zn-TCPP nanosheets were detached from MOF-CT/PVA, this superhydrophilicity was attributed to the roughness generated by the longitudinal growth of MOF based on Wenzel's model[27].

## Ultrafast water vapor harvesting performance

These characterization results revealed the substantial growth of vertically aligned MOF nanosheets on the hydrogel membrane, creating a surface with superhydrophilicity. We further evaluated the wetting process of sprayed water droplets using high-speed camera to investigate the effects of MOF layer and vertically aligned morphology (Supplementary Fig. 13). When water droplets randomly encountered

the MOF-CT/PVA membrane, the wetting spot was immediately visible, and it rapidly spread in 0.02 s (Supplementary Fig. 14). The entire membrane was almost completely wetted after the 0.08 s, indicating that the encountered droplets were directly captured and rapidly absorbed into the underlying hydrogel substrate. As a comparison, even with an extended observation time of 0.9 s, the CT/PVA membrane could not achieve complete wetting. The attached droplets of HMOF-CT/PVA exhibited a tendency for horizontal wetting, resulting in the formation of a larger wetting surface in limited sections. However, the adsorption kinetics were relatively slower compared to MOF-CT/PVA, highlighting the significant influence of vertically aligned morphology on water droplet transfer.

The water vapor harvesting performance of membrane harvester is significantly influenced by water vapor condensation and droplet transportation processes. As illustrated in Fig. 3a and Supplementary Fig. 15, the moisture was firstly condensed on MOF nanosheets based on the capillary condensation caused by superhydrophilic porous structures (Supplementary Fig. 9). Then, the condensed water was transferred to the CT/PVA hydrogel substrate for storage through MOF nanosheets or pores channels between MOF nanosheets. In contrast to a bulk morphology (Supplementary Fig. 16), we assume that the faveolate macropores in the MOF layer provide rapid transport channels with low tortuosity, which further enhances its ability to capture water vapor. Figure 3a illustrates the schematic mechanism of water vapor capture and water transportation of CT/PVA, HMOF-CT/PVA, and MOF-CT/PVA. To further understand the transportation mechanism between MOF nanosheets, computational fluid dynamics simulations were conducted to simulate the transport process of water droplets that are continuously attached to the surface or pores of the MOF nanosheets at higher RH. Simulations were performed (Fig. 3b) to investigate the adsorption processes of both horizontally and vertically distributed structures of the MOF layer within a simplified two-dimensional flow field. Despite the MOF nanosheets' considerable wettability, water droplets captured in the horizontal distribution structure need to pass through or bypass multiple gas−solid interfaces, which significantly reduces the mass transfer rate. At the simulation of 100 ns, the gas−liquid interface of the horizontal nanosheets layer slightly changed in the vertical direction, whereas water molecules completely reached the hydrogel substrate in the vertical MOF nanosheet channels. Moreover, the small pores formed by the vertical distribution of nanosheets are more favorable for water droplet entry in the initial stage based on mass transfer kinetics fitting when compared to larger-sized pores (Supplementary Fig. 17). The difference in Laplace pressure between regions of high (edges and small wedges of nanosheets) and low (thicker part and large pores between nanosheets) curvature further promotes rapid directional adsorption towards the hydrogel substrate.

Water vapor sorption tests under constant temperature and humidified airflow confirmed this hypothesis (Fig. 3c). To comprehensively assess the impact of MOF layer, a thin CT/PVA film with a casting thickness of 100 μm was prepared (Supplementary Table 5). Membranes with other thicknesses are presented in Supplementary Fig. 18. Without MOF coating, the substrate hydrogel membrane exhibits a steady saturation of water vapor adsorption for over 60 min with limited water uptake (2.08 g g$^{-1}$) under 90% RH. The addition of an HMOF layer with a high affinity for water vapor enhances the adsorption rate and overall water uptake to 2.67 g g$^{-1}$, while the vertically aligned MOF morphology boosts water uptake to 4.44 g g$^{-1}$ at 90% RH, reaching 91.4% saturation within 15 min. As the same mass of MOF layer was coated on the same CT/PVA substrate in HMOF-CT/PVA and MOF-CT/PVA, we attribute this rapid water vapor capture to the directional transportation of vertically aligned nanostructure.

Moreover, the controlled morphology of the MOF nanosheets led to an increase in overall water uptake from 30% RH to 90% RH, surpassing that of CT/PVA and HMOF-CT/PVA (Fig. 3d and

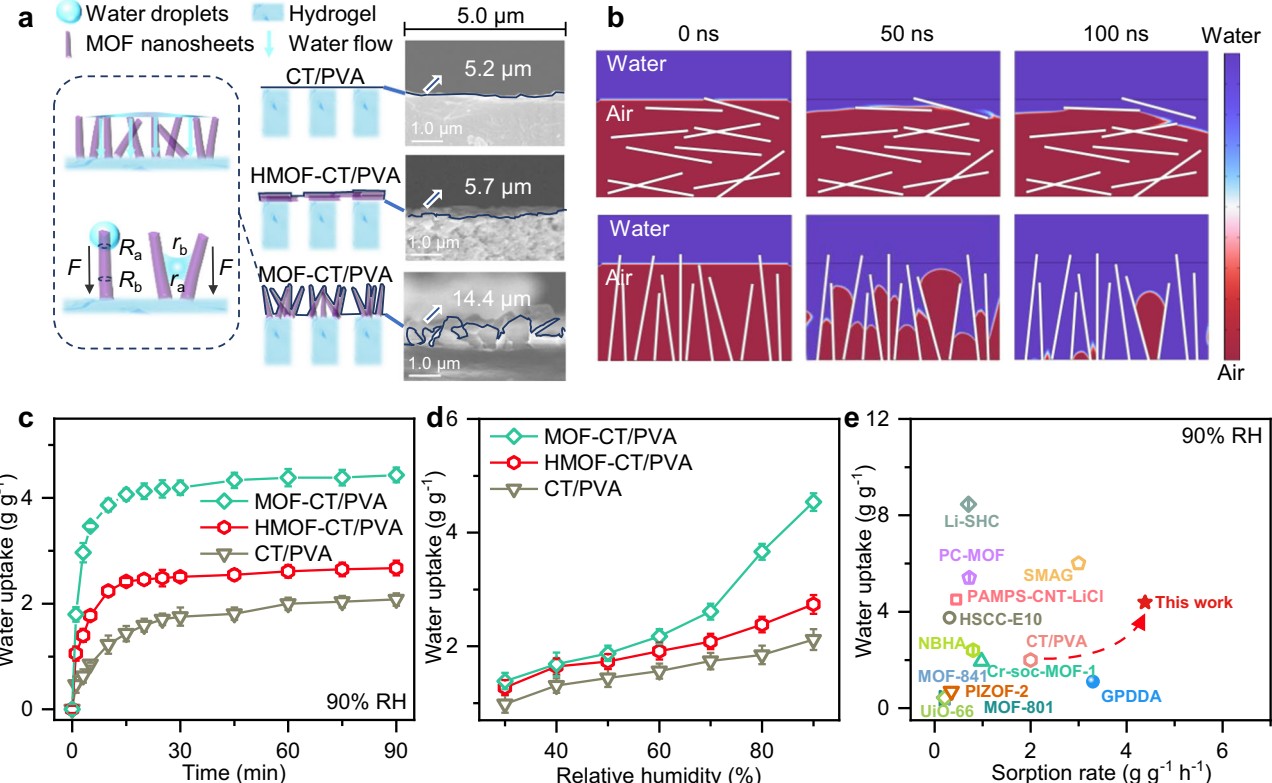

**Fig. 3 | AWH performance of MOF-CT/PVA and morphology effect on water transportation. a** Schematic of water droplets transport channels and sorbent-air interfaces of CT/PVA, HMOF-CT/PVA, and MOF-CT/PVA. The vertical alignment of MOF nanosheets enhances the water capture and transportation capabilities of MOF-CT/PVA (highlighted within dark blue dash frames). $R_a$ and $R_b$ are the local radii of the nanosheets. $r_a$ and $r_b$ represent the transverse radius at the two opposite menisci. $F$ is the Laplace pressure difference. Inset: the sorbent-air interface edges were outlined by solid dark blue lines in the cross-sectional SEM images of CT/PVA, HMOF-CT/PVA, and MOF-CT/PVA. **b** Two-phase flow simulation of water transportation behavior on horizontally and vertically distributed structures of the MOF layer based on computational fluid dynamics. **c** The water vapor collection performance of CT/PVA, HMOF-CT/PVA, and MOF-CT/PVA at 90% RH. The error bars represent the standard deviation of data from three distinct samples ($n = 3$). **d** The water vapor collection performance of CT/PVA, HMOF-CT/PVA, and MOF-CT/PVA at different RH. The error bars represent the standard deviation of data from three distinct samples ($n = 3$). **e** Comparison of water uptake (g g$^{-1}$) and sorption rate (g g$^{-1}$ h$^{-1}$) for AWH between MOF-CT/PVA and sorbents reported in the literature. The AWH performance data are shown in Supplementary Table 4.

Supplementary Fig. 19). Even at a relatively low RH of 30%, MOF-CT/PVA had a water uptake of 1.21 g g$^{-1}$ within 30 min. This difference became more pronounced at RH levels exceeding 70%, attributed to the expanded sorbent-air interfaces. The calculations presented in Fig. 3a indicated a significant increase of 14.4 μm in the boundary contour of MOF-CT/PVA in a two-dimensional plane, while HMOF-CT/PVA exhibited a limited increase from 5.2 μm to 5.7 μm in the same length cross-section. This expanded nanosheet edge, characterized by a vertically aligned morphology, reflected a considerably increased sorbent-air interface, leading to higher water uptake and adsorption rate. Besides, SEM analysis (Fig. 3a) demonstrated that the exposed MOF nanosheets significantly enlarged the windward area, thereby effectively increasing the chances of water vapor capture under constant humidified airflow[28]. It is important to note that the shrinkage rate of the droplet surface area is much lower at higher RH levels, resulting in a slow decrease in the diameter of condensed water droplets and a longer time for them to grow and coalesce into larger water droplets. This further amplifies the disparity in water vapor capture ability between the membrane harvesters discussed in this study at the same time scale. In comparison to previously reported unitary MOF, hydrogel, and other state-of-the-art water harvesters (Fig. 3e and Supplementary Table 4)[4,9,10,29–34], MOF-CT/PVA developed in this study combines the rapid kinetics of MOF materials with the high water uptake of hydrogel, and further enhanced its water vapor capture ability through the incorporation of a vertically aligned morphology.

## Solar-driven evaporation and cycling performance

To harvest the absorbed water in MOF-CT/PVA, our design of vertically aligned MOF layer water harvester is endowed with solar-driven water release capabilities based on the solar-thermal properties of Zn-TCPP and morphology regulation. The photothermal properties of MOF-CT/PVA are assessed by monitoring the surface temperature with an infrared camera. With the expanded solar exposure surface of vertically aligned nanostructure, as a result, the surface can reach equilibrium temperatures to 56.9 °C within 10 min under 1 sun illumination (Fig. 4a). Compared with reported solar-driven membrane layers[33,35], this favorable photothermal ability, combined with surface free energy difference between MOF layer and hydrogel membrane[36], enables efficient solar-driven water release. The sorption and solar-driven desorption process demonstrate that 97% of the absorbed water was released within 10 min under 1 sun, indicating the feasibility of water desorption driven by natural sunlight (Fig. 4b). Therefore, we conducted a continuous cycling sorption–desorption evaluation with 30 min capturing and 10 min releasing, respectively. MOF-CT/PVA showed a stable performance over the tested 10 cycles (Fig. 4c). The average water vapor collection of MOF-CT/PVA is 4.18 g g$^{-1}$ with an average of 97.4% desorption rate, indicating the structural stability and durability of the designed membrane water harvester.

To further analyze the transport mechanism, we supplemented the desorption and cycle process of CT/PVA and HMOF-CT/PVA. The constructed structure of MOF-CT/PVA is beneficial to solar-driven evaporation performance in two ways: (i) The vertically aligned

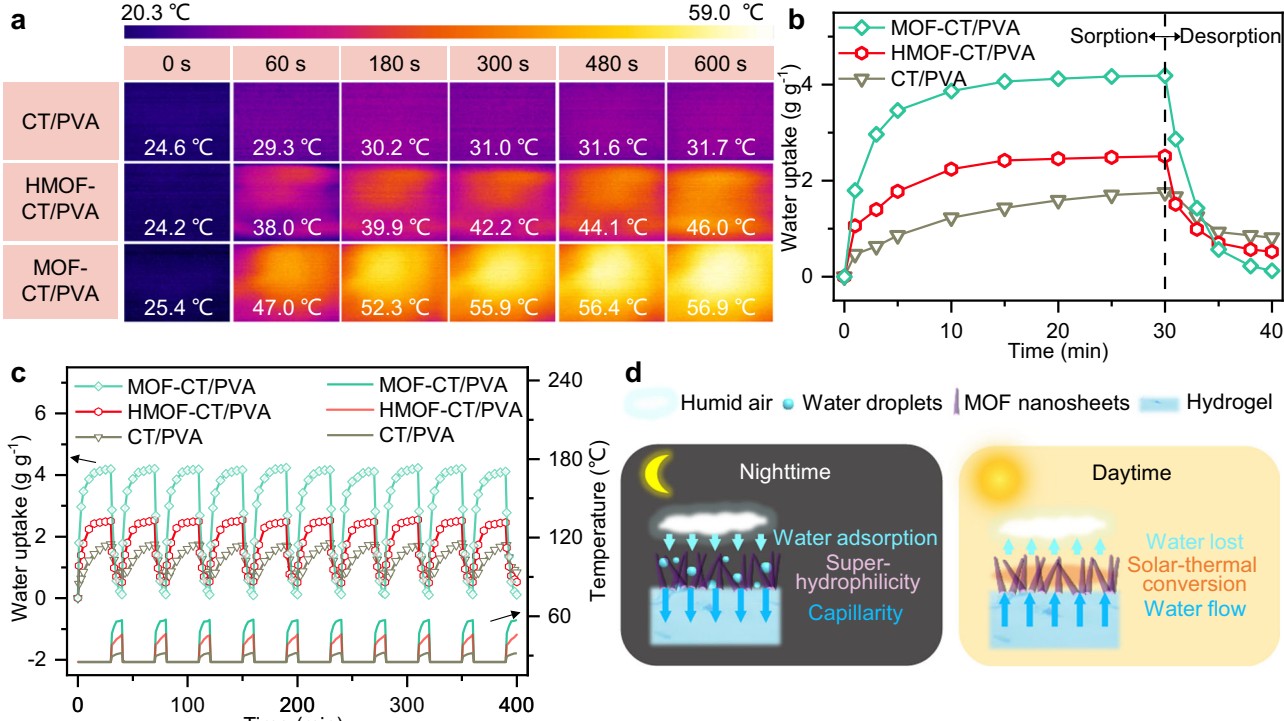

**Fig. 4 | Solar-driven water vapor desorption and cycling performance of hydrogel membrane harvesters. a** Time-dependent surface temperature and the infrared thermal images of CT/PVA, HMOF-CT/PVA, and MOF-CT/PVA under 1 sun illumination. **b** Water vapor sorption curve at 90% RH and desorption curve under 1 sun illumination. **c** Cycling performance and surface temperature of CT/PVA, HMOF-CT/PVA, and MOF-CT/PVA. For water vapor sorption process, membrane samples were placed in a confined climate chamber at 25 °C, which was presented as surface temperature in this figure. **d** Schematic of water adsorption–desorption process under nighttime and daytime environment.

morphology increases the exposed specific surface area to illumination, preventing direct layer-by-layer stacking and aggregation of nanosheets. The significant increase in surface temperature greatly accelerates evaporation kinetics. (ii) The presence of nanosheets on the hydrogel induces a capillary effect on the water absorbed in the hydrogel (Fig. 4d). This, combined with the water channels created by the petal shape and the increased evaporation area, promotes continuous evaporation of the water source, driven by solar energy. The efficient capture and release performance reinforces the potential for using MOF-CT/PVA in AWH. To facilitate large-scale applications, MOF-CT/PVA-1000 was fabricated with a casted thickness of 1000 μm (Supplementary Table 5). As demonstrated in Supplementary Fig. 20, MOF-CT/PVA-1000 consistently exhibited efficient water harvesting capabilities in both indoor and outdoor environments, showing substantial water uptake.

## Discussion

In summary, by using PEG as both a crosslinking and surfactant agent during the construction process, we achieve directional growth of vertically aligned MOF nanosheets onto a hydrogel membrane, resulting in the incorporation of superhydrophilicity. This fabrication strategy overcomes challenges posed by conflicting solvent systems during composite harvester construction. Our findings suggest that PEG molecules adsorb at the nanocrystals and interfaces of the aqueous and organic phases, promoting the formation of an oriented nanosheet layer on the hydrogel surface.

The association of MOFs and hydrogel to construct the composite harvester allows for a judicious combination of their individual properties, leading to ultrafast AWH with high water uptake (4.44 g g$^{-1}$ at a RH of 90%). We highlight that the vertically aligned morphology of the nanosheet layer enables the formation of directional water transport channels and increases the effective contact surface for water vapor harvesting. Our designed AWH membrane harvester exhibits superior sorption–desorption kinetics and robust stability compared to previously reported harvesters in the literature[2]. The synergetic synthesis strategy presented in this study offers a versatile approach for constructing MOFs onto polymeric membranes for the development of multifunctional advanced materials.

## Methods
### Synthesis of hydrogel substrates
The CT/PVA hydrogel membrane substrate was fabricated using the immersed phase inversion process. Firstly, a membrane casting solution was prepared by dissolving 0.27 g CT, 0.80 g PVA, and 0.07 g PEG in 8.86 mL 1% (v/v) acetic acid solution. The mixture was vigorously stirred for 12 h and maintained at 60 °C to remove bubbles. Subsequently, the casting solution was poured onto a clean non-woven fabric and cast using a 100 μm casting knife. After casting, the stainless plate carrying the casting film was immediately immersed in a solution of sodium hydroxide (1 mol L$^{-1}$) and DI water. The resulting membrane was stored in deionized water overnight to complete the phase inversion process. The obtained membrane is abbreviated as CT/PVA.

### Synthesis of vertically aligned MOF membrane harvesters
The Zn-TCPP nanosheets were directionally fabricated onto the CT/PVA substrate. In a 20 mL capped vial, 13.5 mg Zn(NO$_3$)$_2$·6H$_2$O, 2.4 mg pyrazine, and 72 mg PEG were dissolved in a mixture of DMF and ethanol (V:V = 3:1) totaling 36 mL. Then, 12 mg TCPP dissolved in a mixture of DMF and ethanol (V:V = 3:1) totaling 12 mL was added dropwise under stirring. The CT/PVA hydrogel substrate was immersed into this solution at 80 °C using a constant temperature shaker for 24 h. The resulting purple membrane was washed thrice with ethanol. This membrane harvester is denoted as MOF-CT/PVA.

The characteristics of the fabricated membranes with varying thicknesses are detailed in Supplementary Table 5.

## Water vapor sorption–desorption experiments

Dynamic water vapor sorption experiments were carried out in a constant climate chamber (GDJS-80, Hongda), with a temperature accuracy of ±0.1 °C and an RH accuracy of ±1%. The humidity of the chamber was double-confirmed using a digital hygrometer (IACF-305001, TFA), and the difference in humidity values between the digital display of the constant climate chamber and the hygrometer was less than 1% RH. To ensure consistency with literature recommendations, all membrane harvesters were dehydrated by drying at 120 °C for 12 h before the sorption tests, and a constant temperature of 25 °C was maintained throughout all water adsorption experiments. Following the chamber reaching the desired humidity level (ranging from 30% to 90% RH), the dehydrated membrane harvesters were placed inside the chamber, and the weight change of the samples was measured at regular intervals using an analytical balance.

Water evaporation experiments were conducted under simulated sunlight illumination with a power density of $1 \, kW \, m^{-2}$. All sampled membranes were positioned at an equal distance from the light source, in the same location. The room temperature during solar-driven vapor generation was maintained at 25 °C, and the RH was set at 50%. The surface temperature of the membrane harvesters was monitored using an infrared camera (Ti480 PRO, Fluke), and the weight change of the samples was measured at regular intervals using an analytical balance.

## Stability and cycling performance

Based on the water vapor sorption–desorption results, a capturing period of 30 min followed by a releasing period of 10 min was selected for the cycling test. A total of 10 cycles, equating to 400 min of continuous operation, were conducted. After a 30-min fog collection test at 90% RH, the same sample was exposed to simulated sunlight for 10 min in an open space to determine its solar vapor generation rate. The temperature and RH conditions during the cycling performance were kept constant, in line with the parameters of the sorption–desorption experiments described above. MOF-CT/PVA-1000, with a casted thickness of 1000 μm as shown in Supplementary Table 5, was prepared for the indoor and outdoor water vapor adsorption experiment. The indoor test was conducted in a constant climate chamber at 70% RH and 25 °C, while the outdoor test took place under the open sky at 70% RH and 30 °C.

## Characterization

The morphology of membrane harvester was observed by a field emission scanning electron microscope (FE-SEM, S-4800, Hitachi, Japan) with an accelerating voltage of 5 kV. FTIR spectra were measured to determine chemical groups using a Nicolet iS5 FTIR Spectrometer with iD5 ATR Accessory (Thermo Scientific) with the range of $350–4000 \, cm^{-1}$. The BJH equations were used to calculate the pore size destitution with a Quantachrome SI-MP. XPS spectra were recorded in a Thermo ESCALAB 250XI to determine surface elemental composition and valency. The survey spectra were acquired at a pass energy of 150 eV and energy step size of 1 eV. The zeta potential was measured with Zeta potentiometer (Delsa Nano C, Beckman Coulter). The water and diiodomethane contact angle were measured by using an optical tensiometer (Attension Theta, Biolin Scientific). The wetting process of sprayed water droplets was characterized by a high-speed camera with exposures of 0.002 s. The photothermal properties are assessed by monitoring the surface temperature with an infrared camera (Ti480 PRO, Fluke).

## Data availability

The data supporting the findings of the study are included in the main text, Supplementary Information, and Source Data file. Additional data are available from the corresponding author upon request. Source data are provided with this paper.

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

## Acknowledgements

This work was supported by grants from the Innovation and Technology Fund of the Hong Kong Special Administration Region, China (ITS/263/ 22). We appreciate the staff from the Electron Microscope Unit of The University of Hong Kong for SEM and TEM sample preparation and analysis. Special thanks go to Dr. Na Liu for her professional support in characterizing water vapor sorption–desorption experiments.

## Author contributions

L.Z., R.L., and C.T. conceived the idea and designed the research. L.Z., R.L., M.L., Y.H., L.L., and H.F. performed experiments. S.Z. and H.Z. provided constructive suggestions for results. L.Z., R.L., S.Z., H.Z., M.C., and C.T. contributed to writing the manuscript.

## Competing interests

The authors declare no competing interests.
