## [Transparent Peer Review file · Nature Communications]

Hydrogel-embedded vertically aligned metal-organic framework nanosheet membrane for efficient water harvesting

Corresponding Author: Professor Chuyang Tang

Version 0:

Reviewer comments:

Reviewer #1

(Remarks to the Author)

In this work, the authors synthesized hydrogel-embedded vertically aligned metal organic framework composite membrane for water harvesting with fast kinetics and high capacity at high RH. But the most challenge for air water harvesting is at low RH region. In addition, there following comments need to be careful addressed.

1. The mechanism by which PEG promotes CT/PVA cross-linking and Zn-TCPP-oriented growth needs further explanation. In addition, whether the PEG is washed off by ethanol after the directed growth of MOFs? (If PEG is washed away, the change in interlayer spacing of MOFs is not related to the presence of PEG; if it is not washed away, the contribution of PEG to water adsorption must be explained.)
2. The massive heterocyclic structure of Zn-TCPP creates a steric hindrance effect that prevents water molecules from adsorbing. Please provide the contact angle and water adsorption curve of pure Zn-TCPP to demonstrate that the increase in adsorption capacity is caused by the presence of Zn-TCPP. Whether the performance enhancement is due to the presence of PEG? Please carry a control experiment using PEG@CT/PVA for proof.
3. The MOF/CT/PVA composites membrane with micron thickness was synthesized for AWH. The actual indoor and outdoor water harvesting tests should be carried to verify its feasibility.
4. Please provide these parameters (weight, area, thickness, etc.) of testing sample during the adsorption-desorption experiments.
5. The adsorption kinetics of film and powder with varying thicknesses lack comparability.
6. The comparison with those works in low RH also should be presented.
7. The format of the references needs to be further checked.

Reviewer #2

(Remarks to the Author)

Hydrogel-based membranes with porous structures have been demonstrated to be a promising adsorbent in atmospheric water harvesting (AWH). The incorporation of porous MOF and hydrogel has recently emerged as a potential strategy for efficient AWH at a wide range of relative humidity. This manuscript proposes a novel approach to fabricate vertically aligned MOF nanosheets on a hydrogel membrane harvester (MOF-CT/PVA), enabling ultrafast water harvesting. The controlled growth of MOF nanosheets in a vertical orientation is an innovative and enlightening approach for the design of MOF-membrane harvesters. Additionally, the water transport through the MOF layer was clearly demonstrated through two-phase flow simulation. I recommend that this manuscript be accepted for publication with some revisions.

Line 25 - Line 27: The phrase "of MOF" can be omitted. Therefore, the revised sentence would be: "This construction approach significantly enhances the efficiency of water vapor adsorption, offering a potential solution for the design of composite membrane harvesters to mitigate the freshwater crisis."

Line 46 - Line 53: The necessity of directional growth of MOF on hydrogel was not well demonstrated in the current statement. It is suggested to highly highlight the innovation of the proposed morphology of MOF nanosheets in this paragraph.

Line 74 - Line 76: It should be noted that dissolution may have caused the morphology depicted in Fig. S2, and therefore "degradation" may not be the most accurate term to use. I suggest the authors add critical references or revise the sentence for more accurate demonstration.

Line 89 - Line 91: The statement on the directed arrangement at the interface of CT/PVA membrane substrate should be referenced.

Line 130 - Line 133: The statement on the XRD characterization results of [110] plane should be referenced.

Fig. 2e should be presented in the supporting materials since the water contact angle has already been illustrated in Fig. 2d.

Line 227 - Line 229: The statement on the enlarged windward area and the shrinkage rate of the droplet surface area should be referenced. It is recommended to add necessary references throughout this manuscript to support the statements.

Line 252 - Line 254: It is suggested to compare this equilibrium temperature of 56.9 °C with previous studies. Is this indicating a favorable photothermal ability compared with other nanomaterials?

Reviewer #3

(Remarks to the Author)

[Editorial Note: See attachment at the end of the file]

Version 1:

Reviewer comments:

Reviewer #1

(Remarks to the Author)

The comments have been addressed well. It is acceptable.

Reviewer #2

(Remarks to the Author)

The authors have addressed all the comments and this revised manuscript could be accepted for publication.

Reviewer #3

(Remarks to the Author)

The author has provided a satisfactory response to the reviewer's comments, and the paper has undergone good revisions. The revised manuscript is recommended for publish.

made.

Responses to the reviewers' comments:

Reviewer #1:

In this work, the authors synthesized hydrogel-embedded vertically aligned metal organic framework composite membrane for water harvesting with fast kinetics and high capacity at high RH. But the most challenge for air water harvesting is at low RH region. In addition, there following comments need to be careful addressed.

Reply:

Thank you for your insightful feedback. We recognize the significance of air water harvesting in low relative humidity (RH) environments. Actually, the vertically aligned MOF morphology could enhance the water uptake by the membrane substrate even at 30% RH. While the enhancement in water absorption capacity is less pronounced compared to high humidity conditions, MOF-CT/PVA still functions as an effective water harvesting absorbent with rapid kinetics. In response to your recommendations, we have included comparisons at low RH levels. Detailed responses and revisions based on your suggestions are provided below.

1. The mechanism by which PEG promotes CT/PVA cross-linking and Zn-TCPP-oriented growth needs further explanation. In addition, whether the PEG is washed off by ethanol after the directed growth of MOFs? (If PEG is washed away, the change in interlayer spacing of MOFs is not related to the presence of PEG; if it is not washed away, the contribution of PEG to water adsorption must be explained.

Reply:

1A. Mechanisms on cross-linking between PEG and CT/PVA

As reported in the literature, PEG can crosslink with both CT and PVA under thermal treatment (References ²⁰ and ²¹ in manuscript). Scheme 1 demonstrates the potential crosslinking reaction between these hydrogel precursors. Furthermore, in the

presence of acetic acid as a solvent, ester linkages are easily formed during the crosslinking reaction. Therefore, the increased relative content of C-O/C-N and C=O functional groups after PEG induction suggests a cross-linking reaction in the hydrogel substrate by PEG molecules.

Scheme 1. Crosslinking scheme of PEG and CT/PVA

We have supplemented the relative content of specific element in Supplementary Fig. 1 to further demonstrate the cross-link effect of PEG. Lines 70-74 have been revised as

“The X-ray photoelectron spectroscopy (XPS) spectra confirmed that the chemically crosslinked CT/PVA substrate exhibited an increased relative content of C-O/C-N and C=O functional groups (Supplementary Fig. 1). This observation is consistent with the crosslinking reaction between PEG and CT/PVA.^{20, 21}”

Supplementary Fig. 1. a, XPS C 1s spectrum and b, relative content (%) of CT/PVA hydrogel membranes before and after crosslinking reaction with PEG.

20. Aboomeirah AA, Kabil MF, Azzazy HME-S. Polyvinyl alcohol-chitosan-polyethylene glycol-glycerol incorporated with Peganum harmala loaded in lipid nanocapsules as an elastic nanocomposite surgical sealant to control bleeding. *Int J Biol Macromol*, 135987 (2024).

21. Kulkarni AR, Hukkeri VI, Sung H-W, Liang H-F. A novel method for the synthesis of the PEG-crosslinked chitosan with a pH-independent swelling behavior. *Macromol Biosci* 5, 925-928 (2005).

Furthermore, as shown in Supplementary Fig. 2, the substrate with PEG addition shows an intact structure after exposure to the solvent used for MOF growth (DMF/ethanol), confirming effective crosslinking of the substrate. In contrast, the substrate experienced significant dissolution and pore unzipping without PEG addition.

Supplementary Fig. 2. SEM images of the surface of CT/PVA hydrogel membranes **a**, before and **b**, after crosslinking reaction with PEG under MOF solvent system.

1B. Mechanism on oriented growth of Zn-TCPP

For the oriented growth of Zn-TCPP, PEG acts as a surfactant and binds to the nanocrystals to inhibit the isotropic growth of Zn-TCPP. To demonstrate this, we synthesized Zn-TCPP using the same precursor concentration but without PEG introduction, which resulted in a bulk morphology of Zn-TCPP (Supplementary Fig. 3a). Upon inclusion of PEG molecules, a significant anisotropic growth of Zn-TCPP was observed, leading to a nanosheet morphology (Supplementary Fig. 3b).

Supplementary Fig. 3. Nanoparticle SEM images of **a**, bulk Zn-TCPP and **b**, Zn-TCPP with the presentation of PEG.

To further address the reviewer's comment, we have revised Lines 78-82 as

“In addition, the inclusion of PEG molecules, which possess alternating hydrophobic and hydrophilic groups, serves as non-ionic surfactants (Fig. 1b) to control the anisotropic growth of Zn-TCPP (TCPP = tetrakis(4-carboxyphenyl)porphyrin) by selectively attaching themselves to the nanocrystals ¹⁶. Without PEG, isotropic growth results in the bulk crystal of Zn-TCPP, as shown in Supplementary Fig. 3.”

1C. The presence of PEG after the oriented growth of Zn-TCPP

PEG was not completely washed off after the MOF growth process. Fig. 2c compares the XPS results between non-modified bulk Zn-TCPP and Zn-TCPP nanosheets obtained by detaching them from the MOF-CT/PVA. The results revealed that the nanosheets contained a greater relative content of C-O functional groups, in contrast to the limited content observed in the XPS results of bulk Zn-TCPP. This indicates the presence of PEG following the oriented growth of Zn-TCPP. Similar findings were observed in the FTIR results (Supplementary Fig. 6), where the stretching vibration of the ν (C-O) bond in Zn-TCPP nanosheets was notably increased compared to that for bulk Zn-TCPP.

Fig. 2. Characterization of vertically aligned MOF nanosheets on hydrogel membrane. **a**, FTIR spectra of nanosheet Zn-TCPP, bulk Zn-TCPP, MOF-CT/PVA, and CT/PVA substrate. **c**, The C 1s spectrum of XPS characterization of nanosheet Zn-TCPP, bulk Zn-TCPP, MOF-CT/PVA, and CT/PVA substrate.

Supplementary Fig. 6. FTIR spectra of nanosheet Zn-TCPP, bulk Zn-TCPP.

Furthermore, as discussed in our **Reply #1A**, PEG molecules have contributed to the crosslinking of CT/PVA, and these crosslinked PEG will not be washed away.

Regarding your comments on the contribution of PEG to water adsorption, please refer to the detailed response in our **Reply #2A to the reviewer's Comment #2**.

2. *The massive heterocyclic structure of Zn-TCPP creates a steric hindrance effect that prevents water molecules from adsorbing. Please provide the contact angle and water adsorption curve of pure Zn-TCPP to demonstrate that the increase in adsorption capacity is caused by the presence of Zn-TCPP. Whether the performance enhancement is due to the presence of PEG? Please carry a control experiment using PEG@CT/PVA for proof.*

Reply:

2A. The control experiment of PEG@CT/PVA

Following the reviewer's suggestion, we have fabricated PEG@CT/PVA membranes to resolve PEG's role in water adsorption. The following changes are incorporated in the revision:

Materials and Methods in supporting information

“Preparation of control PEG@CT/PVA membranes

We fabricated an additional membrane, PEG@CT/PVA, to study the role of PEG in water adsorption. Specifically, 72 mg of PEG was dissolved in a mixture of DMF and ethanol (V:V=3:1) to a total volume of 48 mL, mirroring the PEG amount used in the MOF-CT/PVA. The CT/PVA hydrogel membrane substrate was then immersed in this solution at 80 °C and subjected to continuous agitation for 24 hours. The resulting membrane was rinsed three times with ethanol, and this membrane is referred to as PEG@CT/PVA. The water adsorption results are presented in Supplementary Fig. 21.”

We have also supplemented the discussion on the comparison results in Supplementary Fig. 21 as

“Supplementary Fig. 21 shows similar water vapor adsorption behavior for CT/PVA and PEG@CT/PVA, suggesting that PEG alone had limited influence on vapor uptake. In contrast, MOF-CT/PVA had much greater water vapor adsorption, highlighting that the directional growth of MOFs notably boosts water harvesting.”

Supplementary Fig. 21. The water vapor collection performance of CT/PVA, PEG@CT/PVA, and MOF-CT/PVA at 90% RH. Data are presented as mean \pm standard deviation (n = 3).

2B. The contact angle and water adsorption curve of pure Zn-TCPP

For the additional comments regarding pure Zn-TCPP, the reviewer has a good point on the steric hindrance effect of the heterocyclic structure. However, it is important to mention that Zn-TCPP contains hydrophilic functional groups capable of forming hydrogen bonds with water molecules. Moreover, the interlayer spacing ($\sim 9.3 \text{ \AA}$) exceeds the diameter of water molecules, facilitating water adsorption and infiltration (Supplementary Fig. 8). The favorable hydrophilicity of pure Zn-TCPP (bulk morphology) was further characterized and supplemented in Supplementary Fig. 16a, revealing an average water contact angle of 24.7° .

Supplementary Fig. 8. Simulated crystal structure of Zn-TCPP nanosheets. **a**, Structures of $Zn_2(COO)_4$ paddlewheel metal node, TCP ligand, and the constructed Zn-TCPP nanosheet. **b**, The layered structure of Zn-TCPP nanosheets with an interlayer distance of 9.3 \AA at 296 K ¹.

In response to your suggestion, we have included the water adsorption curve of pure Zn-TCPP in Supplementary Fig. 16b.

“As shown in Supplementary Fig. 16b, we tested the water adsorption performance of pure Zn-TCPP. It is worth noting that only bulk Zn-TCPP could be fabricated without PEG as surfactants. The bulk Zn-TCPP powder was compressed into a tablet to prevent dispersion in the controlled climate chamber. Although the specific surface area of bulk Zn-TCPP is relatively lower than that of Zn-TCPP

nanosheet, the water adsorption performance of pure bulk Zn-TCPP exhibited significant water uptake and kinetics (0.39 g/g water uptake within 90 mins), which is comparable to pure MOF (Supplementary Table 4).”

Supplementary Fig. 16. a, Water contact angle and b, water uptake of pure Zn-TCPP with bulk morphology.

3. *The MOF/CT/PVA composites membrane with micron thickness was synthesized for AWH. The actual indoor and outdoor water harvesting tests should be carried to verify its feasibility.*

Reply:

Following the reviewer’s suggestion, we performed indoor and outdoor water harvesting tests using the MOF-CT/PVA composites membrane with a casted thickness of 1000 μm . The conditions of this experiment were supplemented in the methods section in Lines 360-363 as

“MOF-CT/PVA-1000, with a casted thickness of 1000 μm as shown in Supplementary Table 5, was prepared for the indoor and outdoor water vapor adsorption experiment. The indoor test was conducted in a constant climate chamber at 70% relative humidity (RH) and 25 $^{\circ}\text{C}$, while the outdoor test took place under the open sky at 70% RH and 30 $^{\circ}\text{C}$.”

The comparison results have been supplemented in supplementary Fig. 20. The discussion of the actual indoor and outdoor water harvesting performance has been supplemented in Lines 277-281 as

“To facilitate large-scale applications, MOF-CT/PVA-1000 was fabricated with a casted thickness of 1000 μm (Supplementary Table 5). As demonstrated in Supplementary Fig. 20, MOF-CT/PVA-1000 consistently exhibited efficient water harvesting capabilities in both indoor and outdoor environments, showing substantial water uptake.”

Supplementary Fig. 20. The outdoor and indoor water vapor collection performance of MOF-CT/PVA with a casting thickness of 1000 μm . Data are presented as mean \pm standard deviation ($n = 3$).

4. Please provide these parameters (weight, area, thickness, etc.) of testing sample during the adsorption-desorption experiments.

Reply:

Thank you for pointing this out. We have supplemented these parameters of all testing membrane samples in Supplementary Table 5.

Supplementary Table 5. The characteristics of the fabricated membranes with varying thicknesses.

	Casting thickness	Size		Weight
	(μm)	Length (cm)	Width (cm)	(mg)
CT/PVA	100	4.0	3.0	141.0 ± 0.2
HMOF-CT/PVA	100	4.0	3.0	143.0 ± 0.5
MOF-CT/PVA	100	4.0	3.0	143.0 ± 0.5
MOF-CT/PVA-500	500	4.0	3.0	175.0 ± 0.7
MOF-CT/PVA-1000	1000	4.0	3.0	213.0 ± 0.8

The membrane weights are presented as mean \pm standard deviation (n = 3).

5. *The adsorption kinetics of film and powder with varying thicknesses lack comparability.*

Reply:

To address this comment, we have fabricated MOF-CT/PVA with various casting thickness (100 μm , 500 μm , 1000 μm). The characteristics of these membrane have been supplemented in Supplementary Table 5 and revise the description on the methods section in Lines 333- 335 as

“This membrane harvester is denoted as MOF-CT/PVA. The characteristics of the fabricated membranes with varying thicknesses are detailed in Supplementary Table 5.”

The results of the adsorption curve of MOF-CT/PVA with various casting thickness have been supplemented in Supplementary Fig. 18:

“We fabricated MOF-CT/PVA with various casting thickness (100 μm , 500 μm , and 1000 μm). Supplementary Fig. 18 shows that, as the thickness of the membrane substrate increased, the water uptake of MOF-CT/PVA decreased progressively. Despite this trend, MOF-CT/PVA-1000 maintained a high water uptake of 3.31 g/g at 90% RH even with a casting thickness of 1000 μm . It is worth noting that the total amount of water adsorption per membrane area significantly increased with the increase of membrane thickness, with MOF-CT/PVA-1000 achieving 226.2 g/m²

compared to 44.4 g/m² for MOF-CT/PVA (thickness = 100 μm). These results highlight the potential for significant practical applications of the relatively thick membrane MOF-CT/PVA-1000 in water vapor harvesting.”

Supplementary Fig. 18. The water vapor collection performance of MOF-CT/PVA with different casting thickness of 100 μm, 500 μm, and 1000 μm. Data are presented as mean ± standard deviation (n = 3).

Besides, Lines 212-214 have been revised as

“To comprehensively assess the impact of MOF layer, a thin CT/PVA film with a casting thickness of 100 μm was prepared (Supplementary Table 5). Membranes with other thicknesses are presented in Supplementary Fig. 18.”

6. *The comparison with those works in low RH also should be presented.*

Reply:

Thank you for your constructive comments. We have included the water adsorption curve of control and MOF-CT/PVA membranes at low relative humidity (30%, 40%, 50%, and 60%) in Supplementary Fig. 19. In addition, the discussion on the comparison results in low RH has been supplemented in Lines 222-225 as

“Moreover, the controlled morphology of the MOF nanosheets led to an increase in overall water uptake over a wide range of RH (30% RH to 90% RH), surpassing that of CT/PVA and HMOF-CT/PVA (Fig. 3d and Supplementary Fig. 19). Even at a relatively low RH of 30%, MOF-CT/PVA had a water uptake of 1.21 g/g within 30 minutes.”

Supplementary Fig. 19. The water vapor collection performance of CT/PVA, HMOF-CT/PVA, and MOF-CT/PVA at relatively low RH (30%, 40%, 50%, and 60%). Data are presented as mean \pm standard deviation ($n = 3$).

7. *The format of the references needs to be further checked.*

Reply:

Thank you for pointing this out. We have corrected the format of the references in the manuscript.

Reviewer #2:

Hydrogel-based membranes with porous structures have been demonstrated to be a promising adsorbent in atmospheric water harvesting (AWH). The incorporation of porous MOF and hydrogel has recently emerged as a potential strategy for efficient AWH at a wide range of relative humidity. This manuscript proposes a novel approach to fabricate vertically aligned MOF nanosheets on a hydrogel membrane harvester (MOF-CT/PVA), enabling ultrafast water harvesting. The controlled growth of MOF nanosheets in a vertical orientation is an innovative and enlightening approach for the design of MOF-membrane harvesters. Additionally, the water transport through the MOF layer was clearly demonstrated through two-phase flow simulation. I recommend that this manuscript be accepted for publication with some revisions.

Reply:

We appreciate your positive comments on our manuscript. Each of your comments has been addressed in our revision (highlight in yellow in the revised manuscript). Detailed responses and revisions based on your suggestions are listed below.

1. *Line 25 - Line 27: The phrase "of MOF" can be omitted. Therefore, the revised sentence would be: "This construction approach significantly enhances the efficiency of water vapor adsorption, offering a potential solution for the design of composite membrane harvesters to mitigate the freshwater crisis."*

Reply:

As your suggestion, we revised the abstract (Lines 25-26) as

“This construction approach significantly enhances the water vapor adsorption, offering a potential solution for the design of composite MOF-membrane harvesters to mitigate the freshwater crisis.”

2. *Line 46 - Line 53: The necessity of directional growth of MOF on hydrogel was not well demonstrated in the current statement. It is suggested to highly highlight the innovation of the proposed morphology of MOF nanosheets in this paragraph.*

Reply:

We have revised Lines 46-53 as

“Additionally, the exfoliation of MOF nanosheets, which offer hierarchically porous pathways and increased surface area for continuous AWH processes ¹⁶, typically requires ultrasonic treatment followed by random deposition onto the membrane surface ¹⁷. These traditional exfoliation and doping methods often lead to significant bulk MOF residues ¹⁸, disordered packing, and weak binding between the MOFs and substrate membranes ¹⁵. In comparison to the controlled growth of MOF nanosheets, these drawbacks further restrict water uptake and decrease the lifespan of the MOF layer due to longitudinal transmission resistance and potential detachment ¹⁴.”

3. *Line 74 - Line 76: It should be noted that dissolution may have caused the morphology depicted in Fig. S2, and therefore "degradation" may not be the most accurate term to use. I suggest the authors add critical references or revise the sentence for more accurate demonstration.*

Reply:

Thank you for pointing this out. We have revised the term to “dissolution” and add critical reference for this statement in Lines 75-77 as

“In contrast, the pristine substrate (without PEG) showed significant dissolution ²² and the formation of unzipping pores in the solvent environment (DMF and ethanol) of MOF synthesis (Supplementary Fig. 2)”

Reference for this statement:

22. Miller-Chou BA, Koenig JL. A review of polymer dissolution. *Prog Polym Sci* 28, 1223-1270 (2003).

4. *Line 89 - Line 91: The statement on the directed arrangement at the interface of CT/PVA membrane substrate should be referenced.*

Reply:

Thanks for your comments. The references for this statement have been supplemented in Lines 90-92 as

“Furthermore, the presence of PEG in a directed arrangement at the interface of CT/PVA membrane substrate further influences the rectangular shape of Zn-TCPP compared to PEG-assisted Zn-TCPP powder materials²⁴.”

Reference for this statement:

24. Martínez-Negro M, Russo D, Prévost S, Teixeira J, Morsbach S, Landfester K. Poly(ethylene glycol)-based surfactant reduces the conformational change of adsorbed proteins on nanoparticles. *Biomacromolecules* 23, 4282-4288 (2022).

5. *Line 130 - Line 133: The statement on the XRD characterization results of [110] plane should be referenced.*

Reply:

Thanks for your comments. The reference for this statement has been supplemented in Lines 131-134 as

“High-resolution transmission electron microscopy (HR-TEM) images (Supplementary Fig. 6) further demonstrated the surface crystallinity of the Zn-TCPP

nanosheets and exhibited lattice fringes with interplanar spacings measuring 1.32 nm, corresponding to the [110] plane ¹⁶.”

Reference for this statement:

16. Zhao M, et al. Ultrathin 2D metal-organic framework nanosheets. *Adv Mater* 27, 7372-7378 (2015).

6. *Fig. 2e should be presented in the supporting materials since the water contact angle has already been illustrated in Fig. 2d.*

Reply:

Thank you for your valuable feedback. The Fig. 2e was included for the comparison of water contact angle and surface free energy. To address your concern, the corresponding results were demonstrated in Lines 161-165 as

“The surface free energy of the CT/PVA substrate was calculated using the Owens, Wendt, Rabel, and Kaelble method (Supplementary eq. 2), revealing a significant increase upon the application of the MOF coating. This enhancement elevated the surface free energy to nearly 81.4 mN/m, accompanied by a distinct vertically aligned morphology (Fig. 2e).”

Fig. 2. Characterization of vertically aligned MOF nanosheets on hydrogel membrane. e, Surface free energy and water contact angle of MOF-CT/PVA, HMOF-CT/PVA, and CT/PVA substrate.

7. *Line 227 - Line 229: The statement on the enlarged windward area and the shrinkage rate of the droplet surface area should be referenced. It is recommended to add necessary references throughout this manuscript to support the statements.*

Reply:

As your suggestion, the reference for this statement has been supplemented in Lines 231-233 as

“Besides, SEM analysis (Fig. 3a) demonstrated that the exposed MOF nanosheets significantly enlarged the windward area, thereby effectively increasing the chances of water vapor capture under constant humidified airflow²⁸.”

28. Shahrokhian A, Feng J, King H. Surface morphology enhances deposition efficiency in biomimetic, wind-driven fog collection. *J R Soc Interface* 17, 20200038 (2020).

Other necessary references have also been included throughout the manuscript (Ref. [20, 33, 35]) and highlighted in yellow.

Included references:

20. Aboomeirah AA, Kabil MF, Azzazy HME-S. Polyvinyl alcohol-chitosan-polyethylene glycol-glycerol incorporated with Peganum harmala loaded in lipid nanocapsules as an elastic nanocomposite surgical sealant to control bleeding. *Int J Biol Macromol*, 135987 (2024).

33. Wang M, et al. Solar-powered nanostructured biopolymer hygroscopic aerogels for atmospheric water harvesting. *Nano Energy* 80, (2021).

35. Wang J, et al. High-yield and scalable water harvesting of honeycomb hygroscopic polymer driven by natural sunlight. *Cell Rep Phys Sci* 3, (2022).

8. *Line 252 - Line 254: It is suggested to compare this equilibrium temperature of 56.9 °C with previous studies. Is this indicating a favorable photothermal ability compared with other nanomaterials?*

Reply:

Thank you for your constructive comments. Compared with previous reported studies, MOF-CT/PVA performed as a favorable photothermal ability. Based on your suggestion, this comparison and references have been supplemented in Lines 256-261 as

“With the expanded solar exposure surface of vertically aligned nanostructure, as a result, the surface can reach equilibrium temperatures to 56.9 °C within 10 mins under 1 sun illumination (Fig. 4a). Compared with reported solar-driven membrane layers^{33, 35}, this favorable photothermal ability, combined with surface free energy difference between MOF layer and hydrogel membrane³⁶, enables efficient solar-driven water release.”

Reviewer #3:

The directional growth of MOF nanosheets on polymeric substrates has emerged as a highly attractive topic in the fabrication of nanocomposite materials for atmospheric water harvesting (AWH). However, the challenge of solvent conflict cannot be ignored. I am pleased to note that the manuscript presents an innovative strategy for fabricating vertically aligned MOF nanosheets on hydrogel membrane substrates, which enables efficient water vapor adsorption. This construction approach offers a novel avenue for the design of multifunctional MOF-materials. Furthermore, the mechanism interpretation through two-phase flow simulation reveals the ultrafast kinetics of this harvester.

Overall, I highly recommend this manuscript for publication with minor revisions. The authors have made significant contributions to the field of AWH and the paper provides valuable insights into the design and development of MOF-based harvesters.

Reply:

We appreciate your positive evaluation and helpful comments on our research. Detailed response and revisions based on your suggestions are listed below.

1. Introduction:

L50: “Unfortunately, conventional exfoliation methods often result in substantial bulk MOF residues...,” The innovation of directional growth of MOF nanosheets should be further improved in this paragraph. The section on the limitations of previous composite membranes with hydrogel and MOFs would benefit from more details to provide readers with a more comprehensive understanding of the challenges faced by researchers in this area.

Reply:

We fully agree with your comments. We have revised the introduction (Lines 46-53) to emphasize the challenges faced in this area

“Additionally, the exfoliation of MOF nanosheets, which offer hierarchically porous pathways and increased surface area for continuous AWH processes ¹⁶, typically requires ultrasonic treatment followed by random deposition onto the membrane surface ¹⁷. These traditional exfoliation and doping methods often lead to significant bulk MOF residues ¹⁸, disordered packing, and weak binding between the MOFs and substrate membranes ¹⁵. In comparison to the controlled growth of MOF nanosheets, these drawbacks further restrict water uptake and decrease the lifespan of the MOF layer due to longitudinal transmission resistance and potential detachment ¹⁴.”

2. Results and discussion:

L70: “The X-ray photoelectron spectroscopy (XPS) spectra confirmed that the chemically crosslinked CT/PVA substrate exhibited an increased relative content of C=O functional groups” The relative content of C 1s functional groups should be supplemented in the supporting information.

Reply:

To address your concern, the relative content of C 1s functional groups has been supplemented in Supplementary Fig. 1. Lines 70-74 has been changed as

“The X-ray photoelectron spectroscopy (XPS) spectra confirmed that the chemically crosslinked CT/PVA substrate exhibited an increased relative content of C-O/C-N and C=O functional groups (Supplementary Fig. 1). This observation is consistent with the crosslinking reaction between PEG and CT/PVA ^{20, 21}.”

Supplementary Fig. 1. a, XPS C 1s spectrum and **b**, relative content (%) of CT/PVA hydrogel membranes before and after crosslinking reaction with PEG.

3. L155: “Zn-TCPP nanosheets were filtered onto the CT/PVA substrate to create a horizontally stacked MOF membrane (HMOF-CT/PVA) with the same mass fraction. SEM images illustrate the stacking of nanosheets, resulting in the formation of a discontinuous layer with an in-plane orientation (Fig. 2d and Supplementary Fig. 10).” The demonstration in Supplementary Fig. 10 of HMOF-CT/PVA should also be supplemented in the manuscript.

Reply:

Thanks for your constructive comments. We have supplemented the detailed description of HMOF-CT/PVA and changed Lines 155-160 as

“To further compare directional MOF growth and conventional filtration of exfoliated MOF nanosheets, Zn-TCPP nanosheets were filtered onto the CT/PVA substrate to create a horizontally stacked MOF membrane (HMOF-CT/PVA) with the same mass fraction. It is important to note that despite the ultrasonic dispersion of the solution system of HMOF-CT/PVA beforehand, agglomeration and stacking phenomena unavoidably occur during the nanosheet filtration process.”

4. L161: *“This enhancement elevated the surface free energy to nearly 81.43 mN/m, accompanied by a distinct vertically aligned morphology.”Please maintain a consistent number of significant digits in this study. It should be changed to 81.4 NM/m*

Reply:

Thanks for your comment. We have fixed this problem in Lines 164 as

“This enhancement elevated the surface free energy to nearly 81.4 mN/m...”

5. L164: *“whereas MOF-CT/PVA took less than half a second (Supplementary Fig. 11)” It is proposed to add a supplementary video demonstrating the rapid adsorption of MOF-CT/PVA for a better understanding.*

Reply:

Thank you for your comments. We have supplemented a continuous video for the water contact angle test of MOF-CT/PVA in our supplementary materials. Lines 165-168 have been changed as

“It is noteworthy that the water droplet exhibited gradual absorption by the CT/PVA substrate and HMOF-CT/PVA within 60 and 10 seconds, respectively, whereas MOF-CT/PVA took less than a second (Supplementary Fig. 12 and Video 1).”

Title: Hydrogel-embedded Vertically Aligned Metal-Organic Framework Nanosheet Membrane for Efficient Water Harvesting

The directional growth of MOF nanosheets on polymeric substrates has emerged as a highly attractive topic in the fabrication of nanocomposite materials for atmospheric water harvesting (AWH). However, the challenge of solvent conflict cannot be ignored. I am pleased to note that the manuscript presents an innovative strategy for fabricating vertically aligned MOF nanosheets on hydrogel membrane substrates, which enables efficient water vapor adsorption. This construction approach offers a novel avenue for the design of multifunctional MOF-materials. Furthermore, the mechanism interpretation through two-phase flow simulation reveals the ultrafast kinetics of this harvester.

Overall, I highly recommend this manuscript for publication with minor revisions. The authors have made significant contributions to the field of AWH and the paper provides valuable insights into the design and development of MOF-based harvesters.

Introduction:

L50: “Unfortunately, conventional exfoliation methods often result in substantial bulk MOF residues...”

The innovation of directional growth of MOF nanosheets should be further improved in this paragraph.

The section on the limitations of previous composite membranes with hydrogel and MOFs would benefit from more details to provide readers with a more comprehensive understanding of the challenges faced by researchers in this area.

Results and discussion:

L70: “The X-ray photoelectron spectroscopy (XPS) spectra confirmed that the chemically crosslinked CT/PVA substrate exhibited an increased relative content of C=O functional groups”

The relative content of C 1s functional groups should be supplemented in the supporting information.

L155: “Zn-TCPP nanosheets were filtered onto the CT/PVA substrate to create a horizontally stacked MOF membrane (HMOF-CT/PVA) with the same mass fraction. SEM images illustrate the stacking of nanosheets, resulting in the formation of a discontinuous layer with an in-plane orientation (Fig. 2d and Supplementary Fig. 10).”

The demonstration in Supplementary Fig. 10 of HMOF-CT/PVA should also be supplemented in the manuscript.

L161: “This enhancement elevated the surface free energy to nearly 81.43 mN/m, accompanied by a distinct vertically aligned morphology.”

Please maintain a consistent number of significant digits in this study. It should be changed to 81.4 NM/m

L164: “whereas MOF-CT/PVA took less than half a second (Supplementary Fig. 11)”

It is proposed to add a supplementary video demonstrating the rapid adsorption of MOF-CT/PVA for a better understanding.